# SARS-CoV-2 RNA detection on environmental surfaces in COVID-19 wards

**Xuan Zhou**[1], **HuiXiao Fu**[2], **Guiqin Du**[3], **Xiaoyu Wei**[4], **BingBing Zhang**[3], **Tao Zhao**[3]*

**1** Guiyang Center for Disease Control and Prevention, Guiyang City, Guizhou Province, China,
**2** Department of Science and Education, The First People's Hospital of Guiyang, Guiyang City, Guizhou Province, China, **3** Department of Hospital Infection Management, The First People's Hospital of Guiyang, Guiyang City, Guizhou Province, China, **4** Laboratory of Bacterial Infectious Disease of Experimental Center, Guizhou Provincial Center for Disease Control and Prevention, Guiyang City, Guizhou Province, China

* 15186613384@163.com

**Data Availability Statement:** All relevant data are within the paper and its Supporting Information files.

## Abstract

This study monitored the presence of SARS-Cov-2 RNA on environmental surfaces in hospital wards housing patients with mild, severe, and convalescent Coronavirus Disease 2019 (COVID-19), respectively. From 29 October to 4 December 2021, a total of 787 surface samples were randomly collected from a General Ward, Intensive Care Unit, and Convalescent Ward at a designated hospital for COVID-19 patients in China. All of the samples were used for SARS-Cov-2 detection. Descriptive statistics were generated and differences in the positivity rates between the wards were analyzed using Fisher's exact tests, Yates chi-squared tests, and Pearson's chi-squared tests. During the study period, 787 surface samples were collected, among which, 46 were positive for SARS-Cov-2 RNA (5.8%). The positivity rate of the contaminated area in the Intensive Care Unit was higher than that of the General Ward (23.5% vs. 10.4%, P<0.05). The positivity rate of the semi-contaminated area in the Intensive Care Unit (4.5%) was higher than that of the General Ward (1.5%), but this difference was not statistically significant (P>0.05). In the clean area, only one sample was positive in the Intensive Care Unit (0.5%). None of the samples were positive in the Convalescent Ward. These findings reveal that the SARS-Cov-2 RNA environmental pollution in the Intensive Care Unit was more serious than that in the General Ward, while the pollution in the Convalescent Ward was the lowest. Strict disinfection measures, personal protection, and hand hygiene are necessary to limit the spread of SARS-Cov-2.

## Introduction

At the beginning of 2020, Severe Acute Respiratory Syndrome Coronavirus 2 (SARS Cov-2) began to spread globally. As of 28 December 2022, over 650 million confirmed cases and over 6.6 million deaths have been reported globally. SARS Cov-2 can be transmitted by direct contact or by droplets from infected persons [1]. In addition, infection as a result of contact with contaminated objects or exposure to fomites has been reported. In a NEJM study, SARS CoV-2 was found to survive for three hours in the form of atomization and 72 hours on plastic and

**Funding:** ZX received Grand numbers: Zhu Ke Contract [2022]-4-6. Full name of funder: Guiyang Science and Technology Plan Project. The funders had no role in study design, data collection and analysis, decision to publish, or preparation of the manuscript.

**Competing interests:** The authors have declared that no competing interests exist.

stainless steel surfaces [2]. Cai et al. [3]. investigated a cluster of coronavirus disease cases associated with a shopping mall, and found that indirect transmission of the virus occurred, perhaps due to virus contamination of common objects. Together, these reports indicate that environmental COVID-19 pollution is a potential transmission risk.

Evidence suggests that the viral loads of critical patients are the highest, followed by severe patients, while the viral loads of mild patients are the lowest [4]. Liu et al. reported that the mean viral load of severe cases was around 60 times higher than that of mild cases [5]. However, there is currently a lack of evidence of any differences in environmental pollution as a function of COVID-19 disease severity. In the context of indirect virus transmission, comprehensive monitoring of hospital environmental hygiene is crucial to ensuring the quality of hospital infection control.

In order to monitor hospital environmental hygiene and evaluate the differences in environmental pollution as a function of COVID-19 disease severity, this study investigated SARS-CoV-2 RNA on environmental surfaces in hospital wards containing mild, severe, and convalescent COVID-19 patients, respectively.

## Methods

### Participant characteristics

The hospital under study in this research was the Jangjunshan Hospital in Guiyang, China. There are three isolation wards within this hospital in which confirmed COVID-19 patients are hospitalized: Intensive Care Unit (ICU), General Ward (GW), and Convalescent Ward (CW, Fig 1). The CW is specifically used to hospitalize COVID-19 patients who have no obvious symptoms after treatment and are waiting for their COVID-19 test to turn negative. Each isolation ward is separated into three zones: a contaminated area, a semi-contaminated area, and a clean area. The contaminated area is designed for COVID-19 patients and their contaminated items. The clean area is designed for non-contaminated items. The semi-contaminated area is a buffer area between the contaminated area and the clean area. Medical staff and patients have their own dedicated passages. All wards are equipped with a negative pressure system. The airflow direction is from the clean area to the potentially contaminated area and then to the contaminated area.

### Disinfection

The air disinfection in the isolation wards is based on the Management Specification for Air Purification (WS/T 368–2012) standard. Surface contaminants must be removed by cleaning before disinfection. Specifically, 1000 mg/L chlorine containing disinfectant is used to wipe the surfaces of objects in the isolation ward, every eight hours. Further, sewage is pretreated with sodium hypochlorite before discharge.

### Sampling method for SARS-CoV-2 RNA

Environmental surfaces were sampled using swabs with flocked polyester tips moistened with preservation solution after 1 hour of disinfection. An area not exceeding 100 $cm^2$ was rubbed with a moistened swab. The swab was placed in a 15 mL tube with 1.5 mL of Ringer 1/4 solution for transport.

Laboratory confirmation of the virus was performed using real-time reverse transcription polymerase chain reaction (RT-PCR). The fluorescence PCR instrument was purchased from the TianLong company in China (model: Gentier 96E). The full-automatic nucleic acid extractor was from the TianLong company in China (model: GeneRotex 96). The SARS-Cov-2

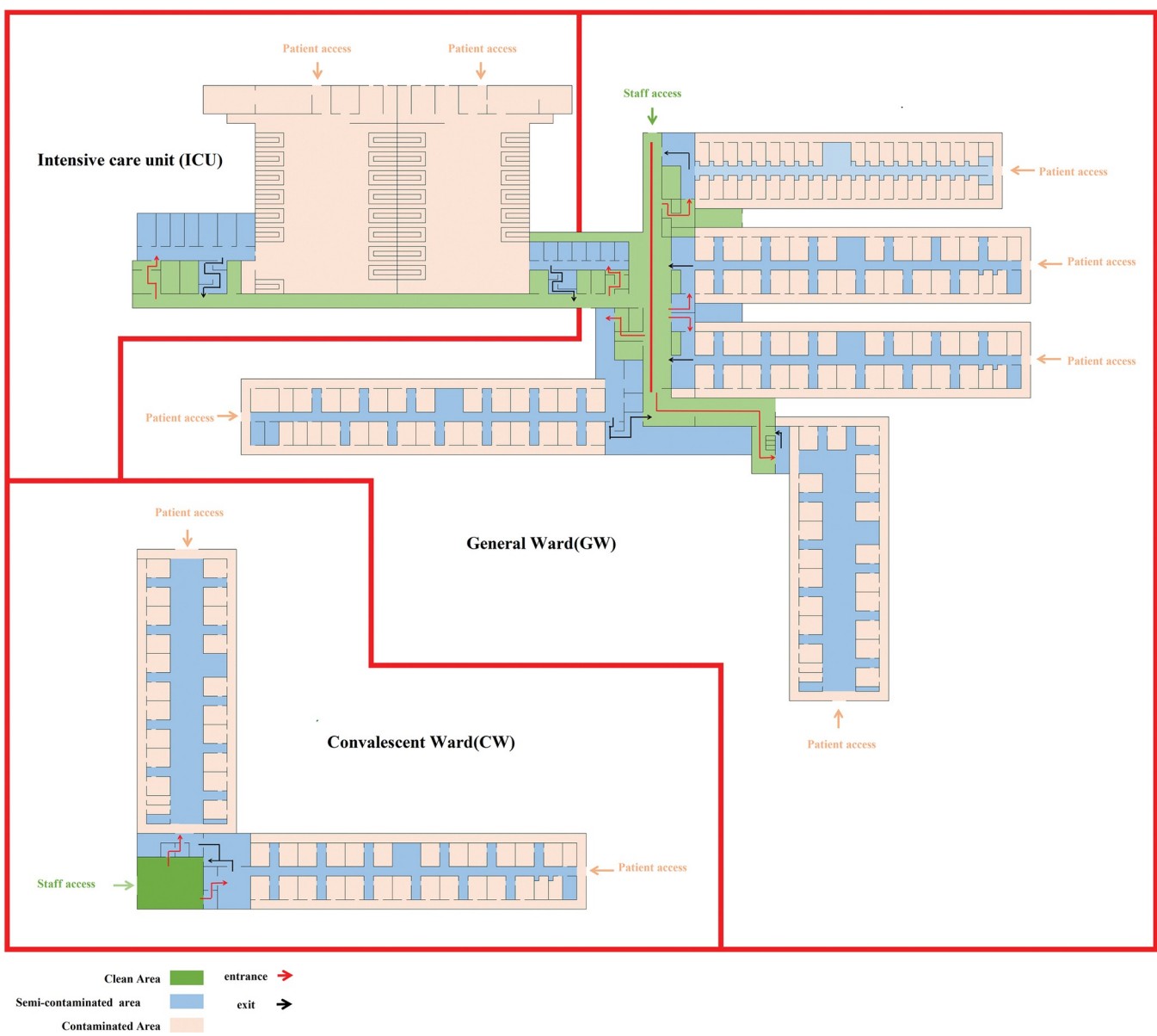

**Fig 1. Partition diagram of an isolation ward.** Color code: Brown = contaminated area; Blue = semi-contaminated area; Green = clean area. Red arrow = Access route; Black arrow = Exit route.

nucleic acid detection kit was from the Da'an gene company of Sun Yat-Sen University, Guangzhou, China. Cycle threshold values were used to quantify the viral load in each sample, with lower values indicating a higher viral load. A sample was considered positive when the qRT-PCR Ct value was <40.

## Statistical analysis

Descriptive statistics were generated and are presented as numbers and percentages. The differences in the positivity rates as a function of COVID-19 severity were compared by Fisher's exact tests, Yates chi-squared tests, and Pearson's chi-squared tests. P<0.05 was taken to indicate statistical significance.

### Ethical approval

This study was approved by the medical ethics committee of the First People's Hospital of Guiyang.

## Results

### ICU

Of the 345 ICU environmental samples, 17 tested positive for SARS-CoV-2 (positivity rate: 4.9%). Among these, 12 samples from the contaminated area tested positive for SARS-CoV-2, including samples taken from the surfaces of medical equipment, door handles, the nurse's station table, and computer keyboards in the nurse's station. In the semi-contaminated area, four samples tested positive for SARS-CoV-2, including samples taken from the dispensing room table, door handles, and hand sanitizer. In the clean area, only one sample (door handle) tested positive (Table 1).

### GW and CW

Of the 372 samples taken from the GW, 29 tested positive for SARS-CoV-2 (positivity rate: 7.8%). Among those taken from the contaminated area, 28 tested positive for SARS-CoV-2, including samples taken from the bed stands, door handles, mobile phones, equipment belts, bed rails, personal protective equipment, medical equipment, infusion stands, and pillows. One sample (door handle) from the semi-contaminated area tested positive for SARS-CoV-2 (Table 2). All samples obtained from the clean area tested negative. All 70 samples obtained from the CW also tested negative (Table 3).

The positivity rate of the ICU contaminated areas (23.5%) was higher than that of the GW (10.4%, $P<0.05$, Table 4). Compared with the GW, the positivity rates of the semi-contaminated area and clean area in the ICU were higher, but these differences were not statistically significant ($P>0.05$, Table 4).

## Discussion

COVID-19 is caused by SARS-CoV-2, which is mainly spread via respiratory droplets.

COVID-19 patients expel the virus through the respiratory tract, causing contamination of the surrounding environment. Previous studies have shown that the survival time of SARS-CoV-2 varies in different environments. For example, studies have reported that the virus can persist for 2 days on glass [6], 1.8 days on stainless steel [7], 7 days on plastic [8], and 7 days on the outer layer of plastic surgical masks [6]. These contaminated surfaces are potential infectious vectors [3]. The hospital environment is often contaminated by SARS-CoV-2, which poses a potential threat to staff. Guo et al [9] detected SARS-CoV-2 RNA in the hospital environment. Futher environmental monitoring assesments are necessary.

Overall, 46 out of 787 swab samples were positive. Seventeen of these were collected in the ICU (4.9%) and twenty-nine in the GW (7.8%) while no viral RNA was found on the surfaces of CW (0.0%), which was in close agreement with the finding of a systematic review by José Gonçalves et al [10].

In the present, the difference of viral RNA among contaminated area, semi-contaminated area and clean area were evaluated. The contaminated area of ICU was at the top position withe the highest virus-positive rate (23.5%, $P<0.05$). This could be related to higher frequency of invasive respiratory droplet-producing operations, such as endotracheal intubation, extubation, bronchoscopy and sputum aspiration, for ICU patients, but not for non-severely ill in GW. Maybe it's also related to severe COVID-19 patients have higher viral loads and longer

**Table 1. SARS-COV-2 RNA results of samples from intensive care unit.**

| Area | Sampling site | Positive samples | Positive rate (%) | Cycle threshold value(1ab/N) |
|---|---|---|---|---|
| **Contaminated area** | Medical equipment | 4/4 | 23.5 | 35.4/34.6 |
| | | | | 35.4/34.6 |
| | | | | 39.7/40.1 |
| | | | | 39.7/40.1 |
| | Spraying sterilizer | 0/2 | | |
| | Equipment belt | 0/26 | | |
| | Transfer window | 0/3 | | |
| | Calling device | 0/1 | | |
| | Door handle | 2/7 | | 35.4/36.4 |
| | | | | 33.709/— |
| | Nurse station stable | 4/4 | | 34.8/35.7 |
| | | | | 35.0/35.6 |
| | | | | 34.8/35.7 |
| | | | | 35.0/35.6 |
| | Computer keyboard of nurse station | 2/2 | | 37.8/37.5 |
| | | | | 37.8/37.8 |
| | Wall surface of decontamination room | 0/2 | | |
| **Semi-contaminated area** | Dispensing room table | 2/2 | 4.5 | 34.3/35.2 |
| | | | | 34.3/35.2 |
| | Door handle | 1/50 | | 39.5/37.7 |
| | Hand sanitizer dispenser | 1/34 | | 37.6/— |
| | Cabinet | 0/2 | | |
| | Wash basin and faucet | 0/1 | | |
| **Clean area** | Door handle | 1/82 | 0.5 | 37.0/39.8 |
| | Consultation room: table, computer keyboard and mouse, landline | 0/44 | | |
| | Transfer window | 0/19 | | |
| | Shoes | 0/12 | | |
| | Cabinet | 0/6 | | |
| | Water dispenser | 0/3 | | |
| | Desk surface | 0/36 | | |
| | Spraying sterilizer | 0/2 | | |
| | Bed | 0/1 | | |
| **Total** | | 17/345 | 4.9 | |

viral shedding periods compared with mild COVID-19 patients [5]. The virus-positive rate in the semi-contaminated and clean areas were lower than contaminated area, but with no statistically significant difference between the ICU and GW ($P>0.05$), which suggests that the pathway separating contaminated area, semi-contaminated ares and clean area as well as the daily cleaning and disinfection protocols were essential.

The highest positivity of viral RNA was found on the surface of medical equipment, door handle, bed rail, personal protective equipment and central nurse station (computer keyboard and mouse), which is consistent with the previous study [11, 12]. These exposed areas and objects were considered to be critical transmission media for SARS-Cov-2 [13].

In our study, one positive door handle sample was collected in clean area of ICU, implying low hand hygiene compliance occasionally. During sampling process in our study, all samples from staffs tested negative for SARS-Cov-2.

**Table 2. SARS-COV-2 RNA results of samples from general ward.**

| Area | Sampling site | Positive samples | Positive rate (%) | Cycle threshold value(1ab/N) |
|---|---|---|---|---|
| **Contaminated area** | Bed stand | 2/2 | 10.4 | 38.7/37.2 |
| | | | | 37.0/37.7 |
| | Door handle | 2/9 | | 37.3/37.9 |
| | | | | 33.4/33.0 |
| | Mobile phone | 1/1 | | 38.4/36.0 |
| | Equipment belt | 3/207 | | 37.2/37.6 |
| | | | | 35.5/37.3 |
| | | | | 34.8/36.5 |
| | Bed rail | 4/11 | | 38.6/38.6 |
| | | | | 33.6/34.6 |
| | | | | 32.8/— |
| | | | | 33.9/— |
| | Intercom | 0/1 | | |
| | Personal protective equipment | 5/8 | | 32.2/— |
| | | | | 37.3/— |
| | | | | 35.5/— |
| | | | | 37.6/— |
| | | | | 39.0/— |
| | Medical equipment | 6/11 | | 37.9/— |
| | | | | 38.3/— |
| | | | | 39.6/— |
| | | | | —/37.7 |
| | | | | 34.6/— |
| | | | | 37.9/36.8 |
| | Table | 0/6 | | |
| | Infusion stand | 1/2 | | 33.6/— |
| | Pillow | 4/10 | | 34.8/— |
| | | | | 34.7/— |
| | | | | 30.9/— |
| | | | | 34.6/— |
| **Semi-contaminated area** | Door handle | 1/13 | 1.5 | 39.0/— |
| | Transfer window | 0/1 | | |
| | Intercom, computer keyboard and mouse, shoes | 0/5 | | |
| | Dispensing room table | 0/3 | | |
| | Wall surface | 0/44 | | |
| **Clean area** | Door handle | 0/21 | 0.0 | |
| | Table | 0/13 | | |
| | Computer keyboard and mouse, shoes, Water dispenser | 0/4 | | |
| **Total** | | 29/372 | 7.8 | |

There are some limitations of this research that should be noted. First, RNA is a marker of virus shedding, but it does not necessarily indicate the presence of viable virus. Second, the air in each ward was not sampled for SARS-CoV-2 RNA detection in this study.

In conclusion, high-contact surfaces in hospitals are more likely to contain detectable SARS-CoV-2 RNA. In addition, COVID-19 patients with different disease severity may have different abilities to contaminate the surrounding environment. SARS-CoV-2 RNA is more easily detected in the ICU than in the GW and CW. Strict cleaning, disinfection, and hand hygiene protocols are essential to reduce the risk of infection.

**Table 3. SARS-COV-2 RNA results of samples from convalescent ward.**

| Area | Sampling site | Positive samples | Positive rate (%) |
|---|---|---|---|
| Contaminated area | Door handle | 0/3 | 0.0 |
| | Table | 0/2 | |
| Semi-contaminated area | Door handle | 0/12 | 0.0 |
| Clean area | Door handle | 0/27 | 0.0 |
| | Table | 0/26 | |
| Total | | 0/70 | 0.00 |

**Table 4. Comparison of SARS-COV-2 RNA results on environmental surface between intensive care unit and general wards.**

| Area | Intensive care unit | | | General wards | | | P value |
|---|---|---|---|---|---|---|---|
| | Positive samples | Total | Positive rate (%) | Positive samples | Total | Positive rate (%) | |
| Contaminated Area | 12 | 51 | 23.5 | 28 | 268 | 10.4 | <0.05[c] |
| Semi-contaminated area | 4 | 89 | 4.5 | 1 | 66 | 1.5 | >0.05[a] |
| Clean Area | 1 | 205 | 0.5 | 0 | 38 | 0.0 | >0.05[b] |
| Total | 17 | 345 | 4.9 | 29 | 372 | 7.8 | >0.05[c] |

a: Yates chi-squared test; b: Fisher's exact probability; c: Pearson's chi-squared test.

## Supporting information

**S1 File.**
(ZIP)

## Author Contributions

**Conceptualization:** Xuan Zhou, Tao Zhao.

**Data curation:** Xuan Zhou, HuiXiao Fu, Guiqin Du, BingBing Zhang.

**Formal analysis:** Xuan Zhou, HuiXiao Fu, Guiqin Du, Xiaoyu Wei, BingBing Zhang, Tao Zhao.

**Funding acquisition:** Xuan Zhou.

**Investigation:** Xuan Zhou, HuiXiao Fu, Guiqin Du, BingBing Zhang, Tao Zhao.

**Methodology:** Xuan Zhou, HuiXiao Fu, Guiqin Du, Xiaoyu Wei, Tao Zhao.

**Project administration:** Xuan Zhou, Tao Zhao.

**Software:** HuiXiao Fu, Guiqin Du, Xiaoyu Wei, BingBing Zhang.

**Supervision:** Tao Zhao.

**Validation:** Xuan Zhou, Xiaoyu Wei, Tao Zhao.

**Visualization:** HuiXiao Fu, Guiqin Du, Xiaoyu Wei, Tao Zhao.

**Writing – original draft:** Xuan Zhou, HuiXiao Fu, Guiqin Du, Xiaoyu Wei, BingBing Zhang, Tao Zhao.

**Writing – review & editing:** Tao Zhao.

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
