## [Decision Letter · Decision Letter 0]

1 Mar 2023

PONE-D-23-00829SARS-CoV-2 RNA detection on environmental surfaces in COVID-19 wardsPLOS ONE

Dear Dr. Zhao,

Thank you for submitting your manuscript to PLOS ONE. After careful consideration, we feel that it has merit but does not fully meet PLOS ONE’s publication criteria as it currently stands. Therefore, we invite you to submit a revised version of the manuscript that addresses the points raised during the review process.Compact abstract and add some results.Discussion is very poor, improve it.Provide gel pictures as supporting information.provide primer details used throughout workprovide systematic flow-chat of work for clear understanding What is the evidence of samples taken from hospital? The reviewer concerned about the sample collection. Please enlist all the samples taken for analysis from hospital and present their results in tabular form.Moreover, sample taken from surface of all wards immediately after disinfection spray or not?Quantitative real-time reverse transcription 30 PCR (qRT-PCR) methods were used to confirm the presence of SARS-Cov-2 RNA. During analysis DNA amplification graphs generated as well as pictures must be reported to support the results.What is the scenario of COVID test of staff working in represented wards of hospital?The reviewer also wonders about the statistical analysis of the article. It’s better to presents better in tabular form with appropriate tools but tables are missing in submitted manuscript.Please submit your revised manuscript by Apr 15 2023 11:59PM. If you will need more time than this to complete your revisions, please reply to this message or contact the journal office at plosone@plos.org. Please include the following items when submitting your revised manuscript:A rebuttal letter that responds to each point raised by the academic editor and reviewer(s). You should upload this letter as a separate file labeled 'Response to Reviewers'.A marked-up copy of your manuscript that highlights changes made to the original version. You should upload this as a separate file labeled 'Revised Manuscript with Track Changes'.An unmarked version of your revised paper without tracked changes. You should upload this as a separate file labeled 'Manuscript'.

We look forward to receiving your revised manuscript.

Kind regards,

Sadia Ilyas, Ph.D.

Academic Editor

PLOS ONE

Journal Requirements:

5. Please include your tables as part of your main manuscript and remove the individual files. Please note that supplementary tables (should remain/ be uploaded) as separate "supporting information" files

Additional Editor Comments:

Reviewer-1

In the present manuscript, the authors monitor the presence of SARS-Cov-2 RNA among hospital environment surfaces. It is an interesting topic and its findings recommends disinfection measures, personal protection and hygiene practice to lessen the burden of viral infection at hospitals. However, there are certain queries that must be address before taken further process of publication.

1. What is the evidence of samples taken from hospital? The reviewer concerned about the sample collection. Please enlist all the samples taken for analysis from hospital and present their results in tabular form.

2. Moreover, sample taken from surface of all wards immediately after disinfection spray or not?

3. Quantitative real-time reverse transcription 30 PCR (qRT-PCR) methods were used to confirm the presence of SARS-Cov-2 RNA. During analysis DNA amplification graphs generated as well as pictures must be reported to support the results.

4. What is the scenario of COVID test of staff working in represented wards of hospital?

5. The reviewer also wonders about the statistical analysis of the article. It’s better to presents better in tabular form with appropriate tools but tables are missing in submitted manuscript.

6. This reviewer could not understand what new outcome can be drawn from the entire workout. Even without doing this study, all the said conclusions are traditionally known. What is the novelty of this article?

7. At last, according to the reviewer, the content of this manuscript is very poor and has nothing new to know for the readers of this journal. Highlight the novelty of research.

Reviewer-2

This study is interesting yet a kind of confirmatory research; however, the study design and significance are not comprehensible and hard to understand.

Additionally, the writing and referencing is not of a decent quality. Overall, this paper does not meet the stringent quality criteria of PLOS ONE.

Reviewers' comments:

Reviewer's Responses to Questions

**Comments to the Author**

1. Is the manuscript technically sound, and do the data support the conclusions?

Reviewer #1: Partly

Reviewer #2: Partly

2. Has the statistical analysis been performed appropriately and rigorously? 

Reviewer #1: No

Reviewer #2: Yes

3. Have the authors made all data underlying the findings in their manuscript fully available?

Reviewer #1: No

Reviewer #2: No

4. Is the manuscript presented in an intelligible fashion and written in standard English?

Reviewer #1: Yes

Reviewer #2: No

5. Review Comments to the Author

Reviewer #1: In the present manuscript, the authors monitor the presence of SARS-Cov-2 RNA among hospital environment surfaces. It is an interesting topic and its findings recommends disinfection measures, personal protection and hygiene practice to lessen the burden of viral infection at hospitals. However, there are certain queries that must be address before taken further process of publication.

1. What is the evidence of samples taken from hospital? The reviewer concerned about the sample collection. Please enlist all the samples taken for analysis from hospital and present their results in tabular form.

2. Moreover, sample taken from surface of all wards immediately after disinfection spray or not?

3. Quantitative real-time reverse transcription 30 PCR (qRT-PCR) methods were used to confirm the presence of SARS-Cov-2 RNA. During analysis DNA amplification graphs generated as well as pictures must be reported to support the results.

4. What is the scenario of COVID test of staff working in represented wards of hospital?

5. The reviewer also wonders about the statistical analysis of the article. It’s better to presents better in tabular form with appropriate tools but tables are missing in submitted manuscript.

6. This reviewer could not understand what new outcome can be drawn from the entire workout. Even without doing this study, all the said conclusions are traditionally known. What is the novelty of this article?

7. At last, according to the reviewer, the content of this manuscript is very poor and has nothing new to know for the readers of this journal. Highlight the novelty of research.

Reviewer #2: This study is interesting yet a kind of confirmatory research; however, the study design and significance are not comprehensible and hard to understand.

Additionally, the writing and referencing is not of a decent quality. Overall, this paper does not meet the stringent quality criteria of PLOS ONE.

6. PLOS authors have the option to publish the peer review history of their article (what does this mean?). If published, this will include your full peer review and any attached files.

Reviewer #1: No

Reviewer #2: No

---

## [Author Response · Author response to Decision Letter 0]

27 Apr 2023

Dear editor

We feel great thanks for your professional review work on our article. As you are concerned, there are several problems that need to be addressed. According to your nice suggestions, we have made extensive corrections to our previous draft, the detailed corrections are listed below.

No 1. Compact abstract and add some results.

We have compacted abstract and added some results in the manuscript.

No 2. Discussion is very poor, improve it.

We have revised the discussion section in the manuscript.

No 3. Provide gel pictures as supporting information.

There is no gel picture in our experiment, but we provide DNA amplification graphs as supporting information. 

No 4. Provide primer details used throughout work.

We provided photos of working in the file labeled ‘workflow’.

No 5. Provide systematic flow-chat of work for clear understanding.

We provided systematic flow-chat of work in the file labeled ‘workflow’.

No 6. What is the evidence of samples taken from hospital? The reviewer concerned about the sample collection. Please enlist all the samples taken for analysis from hospital and present their results in tabular form.

We enlisted all samples taken for analysis from hospital in file labeled ‘Sampling result statistics tabular form’.

No 7. Moreover, sample taken from surface of all wards immediately after disinfection spray or not?

All of our samples were taken one hour after disinfection.

No 8. Quantitative real-time reverse transcription 30 PCR (qRT-PCR) methods were used to confirm the presence of SARS-Cov-2 RNA. During analysis DNA amplification graphs generated as well as pictures must be reported to support the results.

We provided DNA amplification graphs in the file labeled ‘DNA amplification graphs’ to support the results.

No 9. What is the scenario of COVID test of staff working in represented wards of hospital?

All the scenario of COVID test of staff working in wards of hospital were negative.

No10. The reviewer also wonders about the statistical analysis of the article. It’s better to presents better in tabular form with appropriate tools but tables are missing in submitted manuscript.

We carefully reviewed our data and found some statistical errors. We corrected these statistical errors and provided the original results generated by SPSS in the file labeled ‘Statistical results’.

We deeply appreciate your consideration of our manuscript, and we look forward to receiving comments from the reviewers. If you have any queries, please don’t hesitate to contact me at the address below.

Thank you and best regards.

Yours sincerely,

Tao Zhao

Corresponding author:

Name: Tao Zhao E-mail: 15186613384@163.com

---

## [Editor Report · Decision Letter 1]

10 May 2023

SARS-CoV-2 RNA detection on environmental surfaces in COVID-19 wards

PONE-D-23-00829R1

Dear Dr. Tao Zhao,

We’re pleased to inform you that your manuscript has been judged scientifically suitable for publication and will be formally accepted for publication once it meets all outstanding technical requirements.

Kind regards,

Sadia Ilyas, Ph.D.

Academic Editor

PLOS ONE
---

## [Editor Report · Acceptance letter]

17 May 2023

PONE-D-23-00829R1 

SARS-CoV-2 RNA detection on environmental surfaces in COVID-19 wards 

Dear Dr. Zhao:

I'm pleased to inform you that your manuscript has been deemed suitable for publication in PLOS ONE. Congratulations! Your manuscript is now with our production department. 

Kind regards, 

on behalf of

Prof. Sadia Ilyas 

Academic Editor

PLOS ONE